# EMU: Efficient Negative Sample Generation for Knowledge Graph Link Prediction

## Abstract

Knowledge graph embedding (KGE) models encode information in knowledge graphs for the purpose of predicting new links. In order to effectively train these models, it is essential to learn to discriminate between positive and negative samples. Although prior research has demonstrated that enhancing the quality of negative samples can lead to significant improvements in model accuracy, identifying superior-quality negative samples remains a challenging task. To this end, our paper proposes **E**mbedding **M**utation and **U**nbounded label Smoothing (EMU), a novel approach to generating hard negative samples, in contrast to traditional endeavors aimed at identifying more difficult negatives within the training data. By corrupting the negative samples with mutations derived from true samples, EMU creates more challenging and informative negative samples that are harder to distinguish from true samples. Importantly, EMU's simplicity allows it to be seamlessly integrated with existing KGE models and any negative sampling methods. Our experiments show consistent improvement of link prediction performance with various KGE models and negative sampling methods. An implementation of the method and experiments are available at `https://anonymous.4open.science/r/EMU-KG-6E58`.

## 1 Introduction

Knowledge Graphs (KGs) are graph databases, consisting of a collection of facts about real-world entities that are represented in the form of (head, relation, tail)-triplets. With their logical structure reflecting human knowledge, KGs have proven themselves to be a crucial component of many intelligent systems that tackle complex tasks, such as question answering (Huang et al., 2019), recommender systems (Guo et al., 2022), information extraction (Gashteovski et al., 2020), machine reading (Weissenborn et al., 2018), and natural language processing, such as language modeling (Yang & Mitchell, 2017; Logan et al., 2019), entity linking (Radhakrishnan et al., 2018), and question answering (Saxena et al., 2022). Popular KGs such as Freebase (Bollacker et al., 2008), YAGO (Suchanek et al., 2007), and WordNet (Miller, 1995) have been instrumental in driving advancements in both academic research and industrial applications.

One of the major challenges that KGs face is their incompleteness; there may be numerous factually correct relations between entities in the graph that are not covered. To address this issue, the task of link prediction has emerged as a fundamental research topic, aimed at filling in the missing links between entities in the graph. Among the various approaches to predicting these missing links, Knowledge Graph Embedding (KGE) methods have proven to be particularly effective. KGE methods encode entities and relations information into a low-dimensional embedding vector space, thus enabling link prediction using neural networks (Bordes et al., 2013; Yang et al., 2015; Trouillon et al., 2016; Sun et al., 2019).

Various methods have been developed to improve the accuracy of KGE predictions. For instance, Ruffinelli et al. (2020) showed that using contrastive learning improves the model's prediction accuracy, irrespective of the embedding models used. However, to effectively train a model with contrastive learning, it is essential to prepare hard-negative samples that are sufficiently challenging for the model to avoid penalizing true triplets. Although there has been a significant amount of research into effective negative sampling methods (Bordes et al., 2013; Sun et al., 2019; Zhang et al.,

2019; Ahrabian et al., 2020), finding a powerful yet efficient negative sampling method remains an open problem in the research community.

In this paper, we propose EMU, a novel negative sample generation method for the task of KG link prediction. Instead of searching informative negative samples in the training dataset, EMU creates challenging negative samples for the triples used during the training by mutating them with components extracted from the positive one. By merging the embedding vector components of the samples, EMU can effectively control the cosine similarity of the negative samples to the positive ones, a technique we refer to as "Embedding Mutation" in subsection 3.2. To ensure that the true sample components are not erroneously penalized in the generated negative samples, we also introduce "Unbounded Label Smoothing" as an effective regularization method. Figure 1 shows a schematic diagram that illustrates the proposed method. Importantly, EMU's simplicity allows it to be seamlessly integrated with existing KGE models and any negative sampling methods. Through a comprehensive set of experiments, we demonstrate that EMU yields consistent performance improvements across various models and datasets, underscoring its potential as a powerful tool in the field of link prediction.

In summary, the followings are the contributions of our work:

- We propose EMU, a novel technique for generating hard and informative negative samples in KG link prediction tasks.

- We designed EMU to be easily incorporated into existing KGE models and any negative sampling methods.

- We conducted a comprehensive set of experiments to demonstrate the effectiveness of EMU, which shows that EMU consistently improves the performance across different KGE models, datasets, and various negative sampling methods.

## 2 BACKGROUND

Link prediction is a task that consists of finding new links among entities in a graph by leveraging the existing entities and relations. Given a triple (head, relation, tail), one of the elements is omitted (e.g., (head, relation, ?)), and the model is required to predict the missing element to create a new correct triple [1] KGE models have proven to be effective methods for this task because they learn to represent the knowledge of a given graph in a vector space. During training, KGE methods employ negative sampling techniques because KGs contain only information on positive links. Thus, appropriate negative samples are crucial for learning the structure of the KG in embedding space. Negative samples correspond to node pairs that are known *not* to be connected, while positive samples refer to node pairs that are known to be connected. By incorporating negative samples during training, KGE models can improve their ability to distinguish between positive and negative samples, leading to better predictions regarding the presence or absence of links in a graph. To generate good quality negative samples, a range of different techniques are commonly employed. The most frequently utilized approach is Uniform Sampling (Bordes et al., 2013), which corrupts positive samples by substituting the head or the tail of the triple with a uniformly sampled replacement from the KG. However, this technique has limitations since the samples are not sufficiently informative and thus do not enhance training. To address this issue, different negative sampling methods to produce harder samples, or samples that help the model effectively discriminate between positive and negative links, have been proposed, e.g., (Ahrabian et al., 2020).

## 3 EMU

This section contains a comprehensive overview of EMU. We first introduce the notation, followed by an explanation of the mutation mechanism, and conclude with the explanation of unbounded label smoothing.

---

[1] If either "head" or "tail" is omitted, it is denoted as "entity prediction"; If "relation" is omitted, it is denoted as "relation prediction". Although we mainly discussed the "tail" prediction case in the paper for simplicity, our method can also be applied to the other cases.

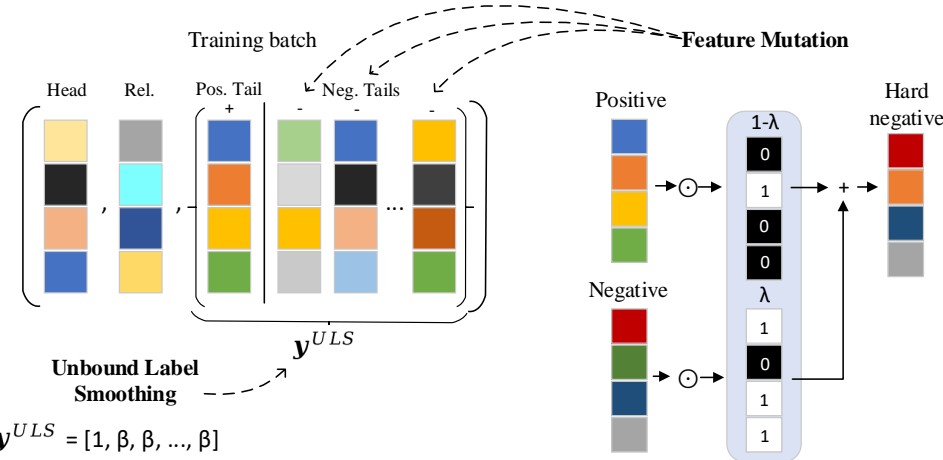

Figure 1: EMU is the combination of the embedding mutation module, and the Unbounded Label Smoothing (ULS). The figure illustrates a typical example that generate hard negative tails.

### 3.1 NOTATION

We introduce the definition of triplets: $x = (\mathbf{h}, \mathbf{r}, \mathbf{t})$ where $(\mathbf{h}, \mathbf{r}, \mathbf{t})$ denote (head, relation, and tail). These triplets are typically large collections of discrete concepts such as ('Joe Biden', 'president of', 'USA'), or ('Tokyo', 'capital of', 'Japan'). Due to their sparsity, the triplets in this format are difficult to handle for machine learning models. Thus, they are usually projected onto a continuous low-dimensional latent space $\mathbf{z} \in \mathbb{R}^d$, where $d$ is the dimensionality of the latent space. The projection is carried out by a projection (embedding) function $G$. For instance, to obtain the projection of the head, we can use the equation $\mathbf{z}_h = G(h|\theta_h)$ where $\theta_h$ is the weight parameter of the embedding model. The inference continues with scoring the triplet feasibility with a scoring function $S(z) = s$ where $z$ is the triple formed by the latents of the input triple $z = (\mathbf{z}_h, \mathbf{z}_r, \mathbf{z}_t)$ and $s$ is the resulting score. Depending on the method, the scoring function $S$ can measure different metrics such as the Euclidean distance as used in TransE (Bordes et al., 2013), or the dot-product as used in DistMult (Yang et al., 2015).

The training is carried out by minimizing a contrastive-type loss function that employs the positive sample score of the true triplet and a set of negative sample scores generated by corrupting the true triple $(+)$ with negative samples $(-)$ as input:

$$\mathcal{L}(s^+, \{s_0, s_1, ...\}^-) \tag{1}$$

The loss function increases the score of the true triple while simultaneously decreasing the score of negative ones. Depending on the method, this can result in a reduction or increase in distances, such as in the case of TransE, or maximizing and minimizing similarities, as in the case of DistMult and others (Trouillon et al., 2016; Sun et al., 2019).

### 3.2 EMBEDDING MUTATION

EMU is inspired from the gene 'mutation' technique utilized in evolutionary algorithms. In this study, we propose a new non-linear mixing approach that replaces a certain amount of the embedding vector components in the negative sample with the corresponding parts of the true positive vector components. This technique is a simple yet effective means of enhancing the difficulty of negative samples by increasing their similarity to the true positive. Figure 2 provides a two-dimensional visualization of this phenomena.

The formal definition of the EMU technique is presented as follows:

$$\tilde{\mathbf{z}}_{\text{EMU}} = \lambda_{\text{EMU}} \odot \mathbf{z}^+ + (1 - \lambda_{\text{EMU}}) \odot \mathbf{z}^-, \tag{2}$$

where $\lambda_{\text{EMU}} \in \mathbb{R}^d$ is the EMU mixing vector that controls the number of embedding vector components to be mutations, which is denoted as $n_{\text{P}}$. $\lambda_{\text{EMU}}$ is a binary-valued vector whose components are generated through a random sampling process that selects either zero or unity, with the probability of $(1 - n_{\text{P}}/d, n_{\text{P}}/d)$. [2] The symbol $\odot$ denotes element-wise multiplication, and $\mathbf{z}^+$ and $\mathbf{z}^-$ correspond to the positive and negative vectors to be mutated, respectively.

For the application of knowledge-base link prediction, we utilize Equation 2 to create the EMU negative tail sample by substituting $z_{t,k} = (\mathbf{z}_t^+, \{\mathbf{z}_{t,0}, \mathbf{z}_{t,1}, \cdots\}^-)$. We employ the generated samples as the EMU negative samples.

### 3.3 Unbounded Label Smoothing

Label smoothing is a well-known technique used to regularize classifier models (Szegedy et al., 2016). It is originally proposed to address the overconfidence issue that certain classifiers such neural networks may exhibit during training. It works by smoothing the class label as follows:

$$\mathbf{y}^{\text{LS}} = \hat{\mathbf{y}}(1 - \beta_{\text{LS}}) + \beta_{\text{LS}}/K, \tag{3}$$

where $\hat{\mathbf{y}} = \{y_0, y_1, ..., y_K\}$ is the *one-hot* label encoding, $\beta_{\text{LS}}$ is a label smoothing parameter that controls the model confidence, and $K$ is the number of classes. Note that the resultant smoothed label maintains the total sum equal to unity. However, when applied to problems with a high number of classes, label smoothing leads to small values for the negative class label (or elements for the contrastive learning case), which still induces an overly strong penalty on the EMU negative samples whose vector component include the true positive sample vector that should not be penalized. To address this issue, we propose a new approach called *Unbounded Label Smoothing* (Unbounded-LS), which is defined as follows:

$$y_k^{\text{ULS}} = \begin{cases} 1 & \text{if} \quad k \in (+) \\ \beta & \quad \text{otherwise,} \end{cases} \tag{4}$$

where $\beta$ is the softening parameter over the negative samples. The above modification of the negative sample labels does not affect the probabilistic interpretation of the model output, as it does not change the model output itself. Our unbounded LS discourages the model from penalizing the negative samples excessively.

### 3.4 Overall Loss Terms

Inspired by knowledge distillation (Hinton et al., 2015), we combine the EMU loss function with the usual Cross-entropy loss function, enabling the model to learn from the vanilla negative samplings (i.e., sampled using the existing methods) as well. The overall loss function is expressed as:

$$\mathcal{L} = \mathcal{L}_{\text{CE}}(s^+, \{s_{0,\text{Mut}}, s_{1,\text{Mut}}, \cdots\}^-, y^{\text{ULS}}) + \alpha \mathcal{L}_{\text{CE}}(s^+, \{s_0, s_1, \cdots\}^-; \hat{y}), \tag{5}$$

where $\mathcal{L}_{\text{CE}}$ is the cross-entropy loss function, $\hat{y}$ and $y^{\text{ULS}}$ are the one-hot and Unbound LS labels, respectively. The numerical coefficient $\alpha$ is utilized for weight balancing between the losses.

## 4 Experiments

In this section, we perform experimental evaluation of EMU for the link prediction problem. To ensure an extensive evaluation, we picked commonly used KG embedding models (ComplEX (Trouillon et al., 2016), DistMult (Yang et al., 2015), TransE (Bordes et al., 2013), and RotatE (Sun et al., 2019)) to test with EMU. Furthermore, we evaluate them on three widely-used knowledge graphs, namely, FB15k-237 (Toutanova & Chen, 2015), WN18RR (Dettmers et al., 2018), and YAGO3-10 (Mahdisoltani et al., 2013).

---

[2]More concretely, the component of the vector $\lambda_{\text{EMU}} \in \mathbb{R}^d$ is composed by $n_{\text{P}}$ unities and $d - n_{\text{P}}$ zeros whose order is randomly determined, e.g., $\{0, 1, 1, 0, 0, 0, \cdots\}$. For simplicity we use the random sampling. The study of the better mutation vector $\lambda_{\text{EMU}}$ is our future work.

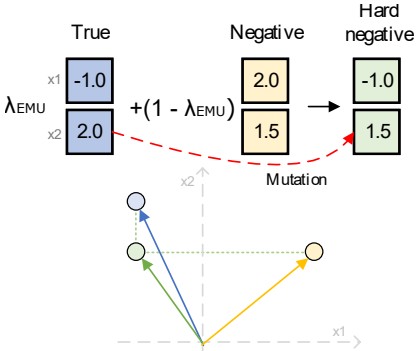

Figure 2: A simple 2D example that illustrates the impact in the latent space of the proposed embedding mutation algorithm.

| Dataset | #entities | #relations | #triples |
|---------|-----------|------------|----------|
| FB15K-237 | 14,541 | 237 | 310,079 |
| YAGO3-10 | 123,188 | 37 | 1,179,040 |
| WN18-RR | 40,943 | 11 | 93,003 |

Table 1: Knowledge Graph dataset statistics. *training*, *validation* and *testing* refer to the number of triples under each split.

## 4.1 DATASETS

**FB15k-237** (Toutanova et al., 2015) is a commonly used benchmark for Knowledge Graph link prediction tasks and a subset of Freebase Knowledge Base (Bollacker et al., 2008). FB15k-237 was created as a replacement for FB15k, a previous benchmark that was widely adapted until the dataset's quality came into question in subsequent work (Toutanova et al., 2015) due to an excess of inverse relations.

**YAGO3-10** (Mahdisoltani et al., 2013) is a subset of YAGO (Yet Another Great Ontology)(Suchanek et al., 2007), a large semantic knowledge base that augments WORDNET and which was derived from WIKIPEDIA (Wikipedia contributors, 2004), WORDNET (Miller, 1995), WIKIDATA (Ahmadi & Papotti, 2021), and other sources. Because of its origins, YAGO entities are linked to WIKIDATA and WORDNET entity types. The dataset contains information about individuals, such as citizenship, gender, profession, as well as other entities such as organizations and cities. The subset YAGO3-10 contains triples with entities that have more than 10 relations.

**WN18RR** (Dettmers et al., 2018) is a link prediction dataset created from WN18 (Bordes et al., 2013), which is a subset of WORDNET, a popular large lexical database of English nouns, verbs, adjectives and adverbs. WORDNET contains information about relations between words, such as `hyponyms`, `hypernyms` and `synonyms` (Miller, 1995). However, similarly to the issues that occurred in FB15K, many test triples in WN18 are obtained by inverting triples from the training set. Therefore, WN18RR dataset was created in the same work as FB15K-237, in order to make a more challenging benchmark for link prediction.

## 4.2 EXPERIMENTAL SETUP

In order to enable a fair comparison between the different models and to ensure that all methods are evaluated under the same conditions, we implemented all the methods. The code to replicate our experiments can be found here: `https://anonymous.4open.science/r/EMU-KG-6E58`.

**Training Settings** Here we describe the general settings we used to train all the models. The optimization was performed using Adam (Kingma & Ba) for $10^5$ iterations[3] with 256 negative samples[4]. The hyper-parameter tuning was performed with Optuna (Akiba et al., 2019). During the training, we monitored the loss over the validation set and selected the best model based on its performance on the validation set. For models trained with SAN negative samples, we utilized the default training setup from (Ahrabian et al., 2020).

---

[3]The total iteration number is the same as the one used in the SAN repository (Ahrabian et al., 2020) to reproduce their best result.

[4]In Appendix D, the influence of the number of negative samples on the outcomes is analyzed, and it is demonstrated that EMU outperforms the uniform-sampling approach in almost all instances.

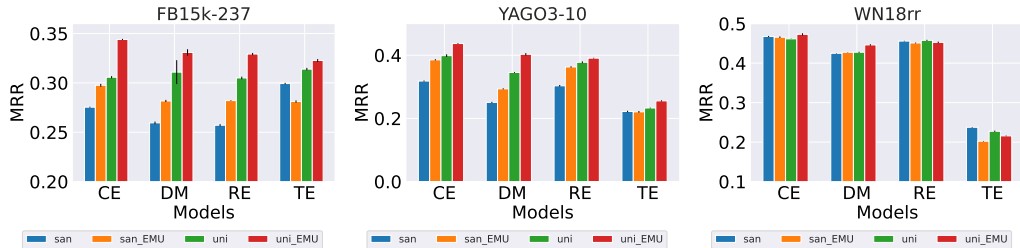

Figure 3: MRR for the datasets: FB15k-237, YAGO3-10, and WN18RR. The blue, orange, green, and red colored bars mean the result of using the following negative sampling methods: "SAN", "SAN with EMU", "uniform", and "uniform with EMU", respectively.

**Evaluation Settings**   To ensure that all the methods were evaluated under the same conditions, we utilized standard metrics to report results, specifically the Mean Reciprocal Rank (MRR) and Hits at K (H@K). If multiple true tails exist for the same (head, relation)-pair, we filtered out the other true triplets at test time. To minimize model uncertainty resulting from random seeds or multi-threading, we performed three trials for each experiment and reported the mean and standard deviation of the evaluation scores.

**Baselines**   Among all existing baselines, we consider vanilla uniform negative sampling (Bordes et al., 2013) and SAN (Ahrabian et al., 2020) as the most relevant to compare our work against. Additionally, We included NSCaching (Zhang et al., 2019) as another baseline method, with its results provided in Appendix C [5]

## 4.3   RESULTS

This section provides a summary and discussion of the obtained results.

Figure 3 illustrates the quantitative results in Table 6, displaying three plots for the MRR results obtained for the FB15k-237, YAGO3-10, and WN18rr datasets. Each plot includes four groups of column bars, representing the results for ComplEX (Trouillon et al., 2016; Lacroix et al., 2018) (CE), DistMult (Yang et al., 2015; Dettmers et al., 2018; Salehi et al., 2018) (DM), RotatE (Sun et al., 2019) (RE), and TransE (Bordes et al., 2013) (TE). The columns are distinguished by colors that correspond to the results obtained from running the SAN (in blue), SAN_EMU (i.e.: SAN negative sampling method with EMU, in orange), uniform sampling (green), and uni_EMU (i.e.: simple uniform negative sampling with EMU, in red). The results indicate that in most cases, the use of EMU enhances the scores by a significant margin, regardless of the embedding model employed. This is discussed in detail in subsection 4.5.

## 4.4   ABLATION STUDY

In this subsection, we present the results of our ablation study to understand the individual contributions of the two main components of EMU, i.e. the Embedding Mutation and the Unbounded-LS. To achieve this goal, we used the FB15k-237 dataset as a reference benchmark and performed a set of experiments by decoupling the embedding mutation from the Unbounded-LS. We trained the KGE models under three different scenarios: 1) the *EMU*, which represents the proposed EMU combining Embedding Mutation and Unbounded-LS; 2) the baseline without Unbounded-LS (*w/t Unbounded-LS*); 3) the baseline without the Embedding Mutation (*w/t Emb.Mut.*), and 4) the case without EMU (*w/t* EMU). The aim of the experiments was to compare the performance reduction by removing one of the components, thereby gaining insights into their relative importance.

The results obtained from the ablation study are presented in Table 2, with the first two columns indicating the target model and the experiment setup, and the last two columns showing the MRR

---

[5]Due to the inherent intricacy involved in assessing the impact of different implementations (specifically, SAN-based and NSCaching-based codes) on performances, we relocated he results obtained with NSCaching to the Appendix.

| Model | Ablation | MRR | HITS@10 |
|-------|----------|-----|---------|
| ComplEX | EMU | **0.344** | **0.532** |
| | w/t Unbounded-LS | 0.252 (**-0.092**) | 0.411 (**-0.121**) |
| | w/t Emb.Mut. | 0.302 (-0.042) | 0.477 (-0.055) |
| | w/t EMU | 0.306 (-0.038) | 0.486 (-0.046) |
| DistMult | EMU | **0.332** | **0.513** |
| | w/t Unbounded-LS | 0.254 (**-0.076**) | 0.415 (**-0.098**) |
| | w/t Emb.Mut. | 0.300 (-0.032) | 0.477 (-0.036) |
| | w/t EMU. | 0.311 (-0.021) | 0.489 (-0.024) |
| RotatE | EMU | **0.329** | **0.514** |
| | w/t Unbounded-LS | 0.236 (**-0.093**) | 0.386 (**-0.128**) |
| | w/t Emb.Mut. | 0.312 (-0.017) | 0.496 (-0.018) |
| | w/t EMU | 0.305 (-0.024) | 0.484 (-0.030) |
| TransE | EMU | **0.323** | **0.503** |
| | w/t Unbounded-LS | 0.260 (**-0.063**) | 0.423 (**-0.080**) |
| | w/t Emb.Mut. | 0.308 (-0.015) | 0.491 (-0.012) |
| | w/t EMU | 0.314 (-0.009) | 0.479 (-0.024) |

Table 2: Ablation study results on FB15K-237. The number in the parentheses are the difference from the "EMU" results. "LS" means Label Smoothing, and "Emb.Mut." means Embedding Mutation.

and HITS@10 results with the performance loss compared to the baseline. Our results consistently demonstrate that Unbounded-LS has a strong impact on all models[6]. This is quite natural because EMU without Unbounded-LS penalizes not only pure negative samples but also true sample embedding because of Embedding Mutation. We also hypothesize that the effectiveness of Unbounded-LS can also stem from its ability to effectively allow large gradient flow values from negative samples. This is attributed to the relatively large negative sample labels (typically larger than 0.1) and the tendency of Embedding Mutation to create harder negatives, which results in larger loss values. The resulting gradients affect both the positive and negative sample components, ultimately leading to an improved representation of their embedding. In conclusion, Embedding Mutation combined with Unbounded-LS consistently (EMU) improves performance of multiple and diverse models.

### 4.5 MUTATION EFFECT

In this subsection, we analyze and discuss the mutation effect in terms of embedding similarity. We use DistMult as a reference model and train it on FB15k-237 and WN18RR datasets. We visualize the embedding vector of the negative tail obtained with EMU to compare it with other negative sampling strategies, i.e., uniform random sampling and SAN negative sampling.

Figure 4 shows the cosine similarity of negative samples provided by the three strategies. The similarity distributions of negative samples produced by the *uniform* and *SAN* methods are quite low, resulting in "easy" negative samples. In contrast, the negative samples generated by EMU exhibits a much larger similarity, indicating that Embedding Mutation generates harder negative samples than the other methods.

Figure 5 depicts the distribution of true-tails and negative tails for two different datasets, namely FB15k-237 and WN18RR, by plotting the first and second PCA components. The left panel of each figure shows the distribution when negative tails are uniformly sampled, while the right panel depicts the distribution using EMU negative-tails. In the Figure 5, the distribution around a true tail is anisotropic for uniform-negative sampling, while EMU negative-tails show an isotropic distribution. We conjecture that this is representative of the distribution of negative samples around a positive tail, because the first and the second PCA components capture crucial information of the high dimensional embedding. Thus, a tighter isotropic distribution translates to harder negatives, while an anisotropic distribution to easier ones. Moreover, the distributions for FB15k-237 are quite

---

[6]We also compared Unbounded-LS and vanilla LS (Szegedy et al., 2016) in Appendix F and found that Unbounded-LS is more effective than the usual LS

varied in comparison to those of WN18RR. This could be an explanation for the higher performance gain when using EMU for FB15k-237 (see Table 6)

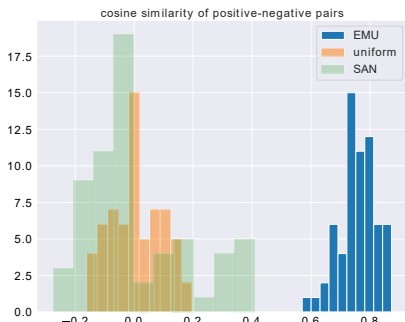

Figure 4: Cosine similarity between positive and negative sample pair for DistMult trained on FB15k-237 dataset. The used negative samples are: uniform, EMU, and SAN. The larger, the more similar.

| Model | Method | MRR | HITS@10 |
|---|---|---|---|
| ComplEX | uni_EMU | **0.344** | **0.532** |
| | uni_MIXUP | 0.324 | 0.517 |
| DistMult | uni_EMU | **0.332** | **0.513** |
| | uni_MIXUP | 0.319 | 0.507 |
| RotatE | uni_EMU | **0.329** | **0.514** |
| | uni_MIXUP | 0.281 | 0.454 |
| TransE | uni_EMU | **0.323** | **0.503** |
| | uni_MIXUP | 0.269 | 0.421 |

Table 3: MRR and Hit@10 of the results on FB15K-237 datasets. "uni_EMU" means the uniform negative sampling with EMU and "uni_MIXUP" means the uniform negative sampling with MIXUP. The shown results are the average with the standard deviation of three trials of the randomly determined initial weights.

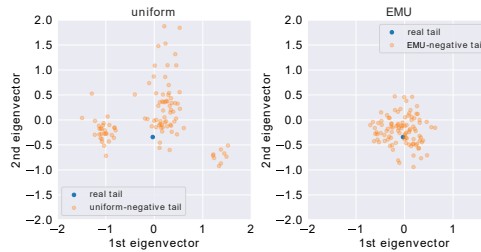
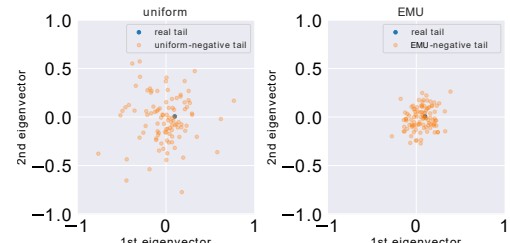

Figure 5: Results of the analysis of EMU of DistMult model trained on FB15k-237 (Left) and WN18RR (Right) dataset. Left: The distribution of real-tail and uniformly-sampled negative-tail in terms of the 1st and 2nd PCA components. Right: The distribution of real-tail and EMU negative-tail in terms of the 1st and 2nd PCA components.

## 4.6 COMPARISON TO MIXUP

In section 5, we discussed a well-known approach called MIXUP, which shares a similar philosophy with the Embedding Mutation. To compare their performance, we replaced the Embedding Mutation step with MIXUP and present the results in Table 3. The findings consistently demonstrate that EMU outperforms MIXUP. We hypothesize that the linear nature of MIXUP-generated examples limits the magnitude of gradients while preserving their direction, thereby restricting its effectiveness. In contrast, EMU overcomes this limitation by generating updates that can explore multiple directions, thereby enhancing model training

## 5 RELATED WORK

**KGE Models** KGE models such as TransE(Bordes et al., 2013), DistMult (Yang et al., 2015; Dettmers et al., 2018; Salehi et al., 2018), ConvE (Dettmers et al., 2018), ComplEX (Trouillon et al., 2016; Lacroix et al., 2018), RotatE (Sun et al., 2019) are commonly used when solving the knowledge base completion task. Each model implements a scoring function which maps a given triple to a real-valued number. These models also differ in the embedding spaces used to learn the latent embedding, for instance RotatE (Sun et al., 2019) utilizes the complex vector space.

**Negative sampling** While training a KGE model for the link prediction task, it is essential to sample high-quality negative data points adequately from the graph. Poor quality negative samples can hinder the performance of KGE models by failing to guide the model during training. With this in

mind, many approaches were proposed for generating better-quality negative samples, i.e., hard negatives. The earliest sampling method is Uniform Sampling (Bordes et al., 2013). Another commonly used method relies on Bernoulli Sampling where the replacement of the heads or tails of the triples follows the Bernoulli distribution. (Wang et al., 2014). Newer methods that are based on Generative Adversarial Networks (GAN) are also used such as KBGAN (Cai & Wang, 2018) and IGAN (Vignaud, 2021) where the generator is adversarially trained for the purpose of providing better quality negative samples where a KGE model is used as the discriminator. Building on this NScasching (Zhang et al., 2019) proposed a distilled version of GAN-based methods by creating custom clusters of candidates entities used for the negative samples. Structure Aware Negative Sampling (SANS) (Ahrabian et al., 2020) leverages the graph structure in the KG by selecting negative samples from a node's k-hop neighborhood. In addition, the subject continues to be actively studied (Zhang et al., 2021; Islam et al., 2022; Xu et al., 2022) . Unlike the prior work mentioned above, EMU generates hard negative samples, distinct from traditional approaches aimed at identifying more difficult negative samples. Furthermore, an additional benefit of EMU is its compatibility with any of the above negative sampling methods, allowing for seamless integration.

**Model Regularization Methods for Classification Tasks**   To obtain a good representation of the embedding vector of a machine learning model, it is common to consider regularization methods. In particular, there have been several regularization techniques for a better generalization power in the case of the cross entropy loss function. MIXUP (Zhang et al., 2018) is one of the most popular and powerful regularization methods, originally developed for image and speech processing. This method generates new training samples by convexly mixing two different training data during the training, resulting in a network with a better generalization because of Vicinal Risk Minimization (Chapelle et al., 2000). Consequently MIXUP has gained popularity in computer vision (Liu et al., 2021; He et al., 2022; Wang et al., 2021) and voice recognition (Meng et al., 2021; Fang et al., 2022), among other fields (Tolstikhin et al., 2021; Kalantidis et al., 2020; Roy et al., 2022; Che et al., 2022). CUTMIX Yun et al. (2019) is a variant of MIXUP that combine two input images as MIXUP but by cutting and pasting patches among images. Our Feature Mutation shares a similar philosophy but the crucial difference is that feature mutation combines positive and negative tails in "feature" space that has not yet been tried in any existing work as far as we know.

Label Smoothing (Szegedy et al., 2016; Müller et al., 2019) is also known as a very effective regularization method when combined with cross entropy loss. Label Smoothing prevents overconfident predictions from the model by artificially reducing the true labels to be less than unity.

## 6 DISCUSSION AND CONCLUSION

In the present study, we proposed our method, EMU, which aims to generate challenging and informative negative samples for knowledge base link prediction by introducing Embedding Mutation and Unbounded Label Smoothing techniques to enhance the embedding model's ability to distinguish true samples. Our comprehensive experimental findings demonstrate that EMU consistently outperforms all the baseline negative sampling methods, including uniform sampling, SAN, and NSCaching in almost all the KGE models and datasets. Moreover, we observed that EMU's efficacy was largely invariant across embedding models and datasets. Our analysis showed that EMU generates negative samples that are closer to true samples in terms of cosine-similarity, and that the generated samples exhibit a more isotropic distribution around the true sample in the embedding space compared to other methods. Although EMU involves tuning a few hyper-parameters, we observed that its performance is not heavily reliant on them (refer to Appendix E). [7].

**Limitations**   MUTUP scope is restricted to KG missing link prediction model trained using the cross-entropy loss function with negative samples. It cannot be applied to neither 1-VS-ALL method nor the other loss functions for the moment.

---

[7](Che et al., 2022) introduced an application of MIXUP to KGE. Their approach involved utilizing more challenging samples for mixing in order to enhance performance. However, it is worth noting that our Unbounded LS technique achieves superior performance compared to their method, even without employing score-based harder sample mixings.

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
