# A EXPERIMENT SETUP

**Training Settings**   We modified the code originally developed by Ahrabian et al. (2020) to perform MIXUP and EMU with SAN. As explained in section 3, the models are trained using cross-entropy losses, incorporating one true tail sample and multiple negative samples. The optimization was performed using Adam (Kingma & Ba). The L3-norm loss function is used on the embedding vectors for the models with the vanilla uniform negative sampling and SAN. The mini-batch size is set to 1000. To compute the embedded triplets For all the KG models, we used an Embedding layer with a hidden dimension of : $d = 100$. A more detailed hyper-parameters are provided in Table 4 and Table 5. We tuned our hyperparameters through 10000 iterations on the FB15K-237 dataset using Optuna (Akiba et al., 2019).

| Model | Method | Learning Rate | $\alpha$ | $n_{\mathrm{P}}/d$ | $\beta$ | $\gamma$ |
|-------|--------|---------------|----------|--------------------|---------|----------|
| ComplEX | uni | 0.1 | n/a | n/a | n/a | $10^{-5}$ |
|         | SAN | 0.1 | n/a | n/a | n/a | $10^{-5}$ |
|         | SAN_EMU | 0.1 | 0.34 | 0.92 | 0.12 | 0 |
|         | uni_EMU | 0.1 | 0.34 | 0.92 | 0.12 | 0 |
| DistMult | uni | 0.1 | n/a | n/a | n/a | $10^{-5}$ |
|          | SAN | 0.1 | n/a | n/a | n/a | $10^{-5}$ |
|          | SAN_EMU | 0.1 | 0.73 | 0.94 | 0.25 | 0 |
|          | uni_EMU | 0.1 | 0.73 | 0.94 | 0.25 | 0 |
| RotatE | uni | 0.005 | n/a | n/a | n/a | $10^{-3}$ |
|        | SAN | 0.005 | n/a | n/a | n/a | $10^{-3}$ |
|        | SAN_EMU | 0.005 | 0.11 | 0.39 | 0.53 | 0 |
|        | uni_EMU | 0.005 | 0.11 | 0.39 | 0.53 | 0 |
| TransE | uni | 0.005 | n/a | n/a | n/a | $10^{-3}$ |
|        | SAN | 0.005 | n/a | n/a | n/a | $10^{-3}$ |
|        | SAN_EMU | 0.005 | 0.11 | 0.39 | 0.53 | 0 |
|        | uni_EMU | 0.005 | 0.11 | 0.39 | 0.53 | 0 |

Table 4: Hyper-Parameters for FB15K-237 and WN18RR dataset. $\alpha, \beta, \gamma$ are the coefficient of EMU Loss, negative label value of Unbounded LS, and the coefficient of L3-norm loss, respectively.

# B A FULL DESCRIPTION OF MAIN RESULT

In Table 6 we provide the full description of our result visualized in Figure 3.

# C RESULTS USING NSCACHING

This section presents the results obtained with EMU and NSCaching (Zhang et al., 2019)[8]. We modified the official NSCaching repository to enable the use of the cross entropy loss function and EMU. We used the same hyperparameters as those provided in section 3, mini-batch size is 1000 and 256 negative samples. EMU parameters are provided in Table 8. The results are provided in Table 7 which demonstrate that our EMU consistently improves the performance, even when using NSCaching[9].

---

[8]We provided the results with NSCaching in the appendix rather than the main body because of differences in the implementation between the official repositories for SAN and NSCaching, which makes it difficult to compare those results equally.

[9]The obtained MMR and H@10 values may appear excessively good; however, we believe that this may be partly due to the NSCaching code implementation, although we cannot confirm this with certainty.

| Model | Method | Learning Rate | $\alpha$ | $n_{\mathrm{P}}/d$ | $\beta$ | $\gamma$ |
|---|---|---|---|---|---|---|
| ComplEX | uni | 0.1 | n/a | n/a | n/a | $10^{-5}$ |
| | SAN | 0.1 | n/a | n/a | n/a | $10^{-5}$ |
| | SAN_EMU | 0.1 | 0.536 | 0.804 | 0.193 | 0 |
| | uni_EMU | 0.1 | 0.536 | 0.804 | 0.193 | 0 |
| DistMult | uni | 0.1 | n/a | n/a | n/a | $10^{-5}$ |
| | SAN | 0.1 | n/a | n/a | n/a | $10^{-5}$ |
| | SAN_EMU | 0.1 | 0.54 | 0.949 | 0.22 | 0 |
| | uni_EMU | 0.1 | 0.54 | 0.949 | 0.22 | 0 |
| RotatE | uni | 0.1 | n/a | n/a | n/a | $10^{-5}$ |
| | SAN | 0.1 | n/a | n/a | n/a | $10^{-5}$ |
| | SAN_EMU | 0.1 | 0.46 | 0.73 | 0.84 | 0 |
| | uni_EMU | 0.1 | 0.46 | 0.73 | 0.84 | 0 |
| TransE | uni | 0.1 | n/a | n/a | n/a | $5 \times 10^{-5}$ |
| | SAN | 0.1 | n/a | n/a | n/a | $5 \times 10^{-5}$ |
| | SAN_EMU | 0.1 | 0.11 | 0.39 | 0.53 | 0 |
| | uni_EMU | 0.1 | 0.11 | 0.39 | 0.53 | 0 |

Table 5: Hyper-Parameters for YAGO3 dataset. $\alpha, \beta, \gamma$ are the coefficient of EMU Loss, negative label value of Unbounded LS, and the coefficient of L3-norm loss, respectively.

| Model | Method | FB15K-237 MRR | HITS@10 | WN18RR MRR | HITS@10 | YAGO3-10 MRR | HITS@10 |
|---|---|---|---|---|---|---|---|
| ComplEX | uni | $0.306^{\pm0.001}$ | $0.486^{\pm0.000}$ | $0.461^{\pm0.000}$ | $0.526^{\pm0.002}$ | $0.399^{\pm0.004}$ | $0.605^{\pm0.003}$ |
| (Trouillon et al., 2016) | SAN | $0.275^{\pm0.000}$ | $0.437^{\pm0.001}$ | $0.467^{\pm0.001}$ | $0.530^{\pm0.001}$ | $0.318^{\pm0.002}$ | $0.496^{\pm0.004}$ |
| | SAN_EMU | $0.298^{\pm0.001}$ | $0.474^{\pm0.001}$ | $0.466^{\pm0.002}$ | $0.543^{\pm0.003}$ | $0.385^{\pm0.002}$ | $0.563^{\pm0.002}$ |
| | uni_EMU | $\mathbf{0.344}^{\pm0.001}$ | $\mathbf{0.532}^{\pm0.001}$ | $\mathbf{0.473}^{\pm0.003}$ | $\mathbf{0.547}^{\pm0.002}$ | $\mathbf{0.437}^{\pm0.001}$ | $\mathbf{0.638}^{\pm0.004}$ |
| DistMult | uni | $0.299^{\pm0.001}$ | $0.476^{\pm0.001}$ | $0.428^{\pm0.001}$ | $0.489^{\pm0.002}$ | $0.345^{\pm0.001}$ | $0.538^{\pm0.004}$ |
| (Yang et al., 2015) | SAN | $0.259^{\pm0.001}$ | $0.415^{\pm0.001}$ | $0.425^{\pm0.001}$ | $0.481^{\pm0.002}$ | $0.251^{\pm0.002}$ | $0.428^{\pm0.001}$ |
| | SAN_EMU | $0.282^{\pm0.001}$ | $0.446^{\pm0.002}$ | $0.427^{\pm0.001}$ | $0.506^{\pm0.004}$ | $0.293^{\pm0.002}$ | $0.478^{\pm0.002}$ |
| | uni_EMU | $\mathbf{0.332}^{\pm0.001}$ | $\mathbf{0.513}^{\pm0.001}$ | $\mathbf{0.446}^{\pm0.002}$ | $\mathbf{0.523}^{\pm0.003}$ | $\mathbf{0.403}^{\pm0.004}$ | $\mathbf{0.601}^{\pm0.004}$ |
| RotatE | uni | $0.305^{\pm0.001}$ | $0.484^{\pm0.001}$ | $\mathbf{0.458}^{\pm0.001}$ | $\mathbf{0.549}^{\pm0.002}$ | $0.378^{\pm0.003}$ | $0.569^{\pm0.003}$ |
| (Sun et al., 2019) | SAN | $0.257^{\pm0.001}$ | $0.418^{\pm0.001}$ | $0.456^{\pm0.001}$ | $0.532^{\pm0.003}$ | $0.303^{\pm0.003}$ | $0.459^{\pm0.003}$ |
| | SAN_EMU | $0.282^{\pm0.000}$ | $0.455^{\pm0.001}$ | $0.451^{\pm0.001}$ | $0.516^{\pm0.002}$ | $0.363^{\pm0.002}$ | $0.535^{\pm0.002}$ |
| | uni_EMU | $\mathbf{0.329}^{\pm0.001}$ | $\mathbf{0.514}^{\pm0.001}$ | $0.453^{\pm0.002}$ | $0.525^{\pm0.002}$ | $\mathbf{0.391}^{\pm0.001}$ | $\mathbf{0.609}^{\pm0.002}$ |
| TransE | uni | $0.314^{\pm0.001}$ | $0.479^{\pm0.002}$ | $0.227^{\pm0.002}$ | $0.506^{\pm0.002}$ | $0.233^{\pm0.001}$ | $0.389^{\pm0.005}$ |
| (Bordes et al., 2013) | SAN | $0.299^{\pm0.001}$ | $0.460^{\pm0.002}$ | $\mathbf{0.237}^{\pm0.001}$ | $\mathbf{0.518}^{\pm0.002}$ | $0.222^{\pm0.002}$ | $0.375^{\pm0.001}$ |
| | SAN_EMU | $0.281^{\pm0.000}$ | $0.450^{\pm0.003}$ | $0.202^{\pm0.001}$ | $0.493^{\pm0.001}$ | $0.221^{\pm0.003}$ | $0.383^{\pm0.001}$ |
| | uni_EMU | $\mathbf{0.323}^{\pm0.001}$ | $\mathbf{0.503}^{\pm0.003}$ | $0.216^{\pm0.001}$ | $0.493^{\pm0.001}$ | $\mathbf{0.255}^{\pm0.002}$ | $\mathbf{0.438}^{\pm0.002}$ |

Table 6: MRR and Hit@10 of the results on FB15K-237, WN18RR, and YAGO3-10 datasets. "uni" means the uniform negative sampling, "SAN" means the structure aware negative sampling. The shown results are the average with the standard deviation of three trials of the randomly determined initial weights.

# D  NEGATIVE SAMPLE NUMBER DEPENDENCE

In the main body of this work, we maintained a fixed number of negative samples at 256. However, in Figure 6, we depict the relationship between the optimal MRR and the number of negative samples employed. Our experiments were conducted using the FB15K-237. Notably, EMU demonstrated superior MRR values in most cases, with a notable increase in performance gains as the number of negative samples increased.

| Model | Method | | MRR | Hit@10 |
|-------|--------|---|-----|--------|
| ComplEX | NSCaching | | $0.387^{\pm 0.001}$ | $0.577^{\pm 0.001}$ |
| | NSCaching_EMU | | $\mathbf{0.394}^{\pm 0.001}$ | $\mathbf{0.585}^{\pm 0.005}$ |
| DistMult | NSCaching | | $0.370^{\pm 0.001}$ | $0.557^{\pm 0.003}$ |
| | NSCaching_EMU | | $\mathbf{0.376}^{\pm 0.002}$ | $\mathbf{0.565}^{\pm 0.000}$ |
| TransE | NSCaching | | $0.322^{\pm 0.001}$ | $\mathbf{0.470}^{\pm 0.002}$ |
| | NSCaching_EMU | | $\mathbf{0.323}^{\pm 0.001}$ | $0.467^{\pm 0.004}$ |
| RotatE | NSCaching | | n/a | n/a |
| | NSCaching_EMU | | n/a | n/a |

Table 7: MRR and Hit@10 of the results with NSCaching code trained using FB15K-237. "NSCaching" means the NSCaching negative smapling. The shown results are the average with the standard deviation of three trials of the randomly determined initial weights. Note that the result of RotatE is omitted because RotatE is not provided in the original repository.

| Model | Method | Learning Rate | $\alpha$ | $n_{\mathrm{P}}/d$ | $\beta$ | $\gamma$ |
|-------|--------|---------------|----------|--------------------|---------|----------|
| ComplEX | uni | $3 \times 10^{-4}$ | n/a | n/a | n/a | $10^{-5}$ |
| | NSCaching | $3 \times 10^{-4}$ | n/a | n/a | n/a | $10^{-5}$ |
| | NSCaching_EMU | $3 \times 10^{-4}$ | 0.44 | 0.34 | 0.32 | 0 |
| DistMult | uni | $10^{-3}$ | n/a | n/a | n/a | $10^{-5}$ |
| | NSCaching | $10^{-3}$ | n/a | n/a | n/a | $10^{-5}$ |
| | NSCaching_EMU | $10^{-3}$ | 0.68 | 0.17 | 0.16 | 0 |
| TransE | uni | $5 \times 10^{-4}$ | n/a | n/a | n/a | $2 \times 10^{-2}$ |
| | NSCaching | $5 \times 10^{-4}$ | n/a | n/a | n/a | $2 \times 10^{-2}$ |
| | NSCaching_EMU | $5 \times 10^{-4}$ | 0.54 | 0.168 | 0.151 | 0 |

Table 8: Hyper-Parameters of NScaching code trained using FB15K-237 dataset. $\alpha, \beta, \gamma$ are the coefficient of EMU Loss, negative label value of Unbounded LS, and the coefficient of L3-norm loss, respectively.

## E HYPER-PARAMETER DEPENDENCE STUDY

In Table 9 illustrates the dependence of EMU performance on hyper-parameters: $\alpha, n_P/d$, and $\beta$. We considered the DistMult as a KGE model. The results indicate that the excessively large values of the coefficient of EMU loss, $\alpha$, are undesirable. Conversely, it is preferable to use a moderate value for the negative label value of Unbounded LS, $\beta$. Finally, the performance is relatively insensitive to the change of the mutation ratio, $n_P/d$, but exhibits a slight improvement as the value approaches the optimal one.

| $(\alpha, n_{\mathrm{P}}/d, \beta)$ | | MRR | HITS@10 |
|-------------------------------------|---|-----|---------|
| $(0.11, 0.914, 0.53)$ **baseline** | | **0.333** | **0.513** |
| $(0.5, 0.914, 0.53)$ | | 0.318 | 0.498 |
| $(0.9, 0.914, 0.53)$ | | 0.306 | 0.484 |
| $(0.11, 0.1, 0.53)$ | | 0.326 | 0.509 |
| $(0.11, 0.5, 0.53)$ | | 0.327 | 0.504 |
| $(0.11, 0.914, 0.1)$ | | 0.314 | 0.496 |
| $(0.11, 0.914, 0.9)$ | | 0.326 | 0.501 |

Table 9: Hyper-Parameters study results using FB15K-237 dataset with DistMult model. $\alpha, \beta, n_P$ are the coefficient of EMU Loss, negative label value of Unbounded LS, and the number of mutation components, respectively.

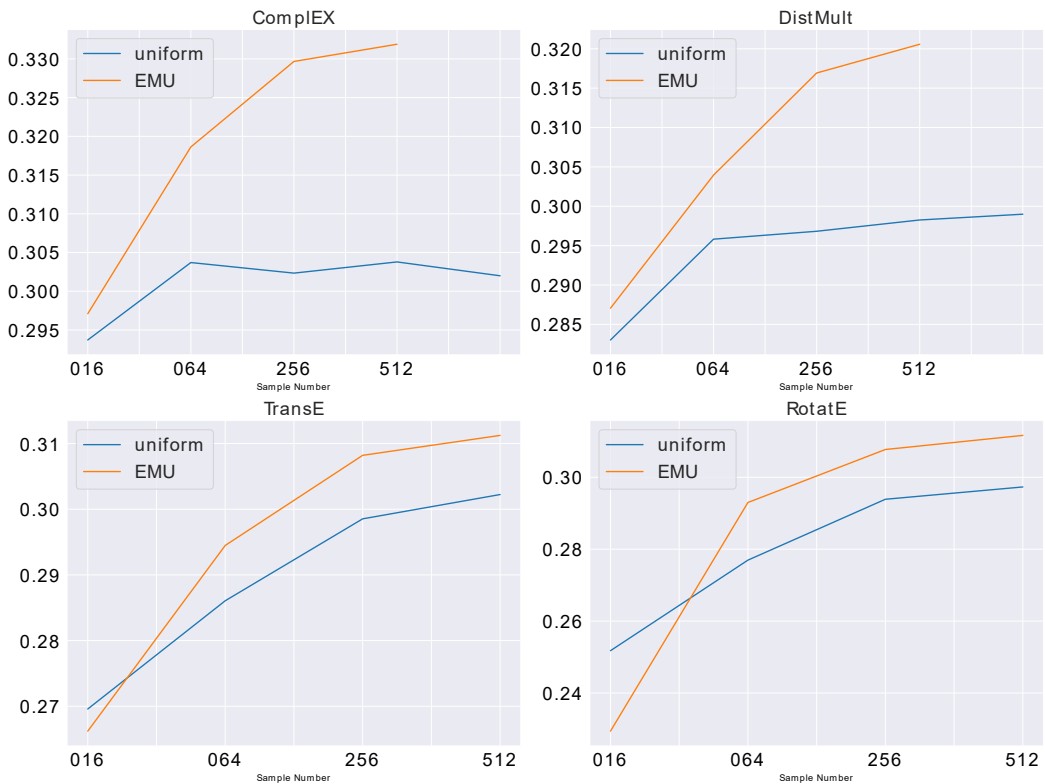

Figure 6: The negative sample number dependence of MRR trained on FB15K-237. The right-edge of the ComplEX and DistMult of the uniform negative sampling case is the "1 VS ALL" results.

## F COMPARISON BETWEEN VANILLA LS AND UNBOUNDED LS

In this study we proposed the unbounded label-smoothing (LS) technique. To assess its efficacy, we also trained our models using vanilla LS (Szegedy et al., 2016) with a label smoothing parameter of 0.2. The result is provided in Table 10 which demonstrate the clear speriority of Unbounded LS for all cases.

| Model | Ablation | MRR | HITS@10 |
|---|---|---|---|
| ComplEX | Unbounded LS | **0.344** | **0.532** |
| (Trouillon et al., 2016) | Vanilla LS (w/t ULS) | 0.262 (**-0.082**) | 0.423 (**-0.109**) |
| DistMult | Unlabeled LS | **0.332** | **0.513** |
| (Yang et al., 2015) | Vanilla LS (w/t ULS) | 0.252 (**-0.080**) | 0.410 (**-0.103**) |
| RotatE | Unbounded LS | **0.329** | **0.514** |
| (Sun et al., 2019) | Vanilla LS (w/t ULS) | 0.236 (**-0.093**) | 0.382 (**-0.132**) |
| TransE | Unbounded LS | **0.322** | **0.503** |
| (Bordes et al., 2013) | Vanilla LS (w/t ULS) | 0.259 (**-0.063**) | 0.423 (**-0.080**) |

Table 10: A comparison between Unbounded LS and vanilla LS.

## G  AN BRIEF INTRODUCTION TO MIXUP

In this section, we provide a brief introduction of *Mixup* (Zhang et al., 2018). MIXUP is a simple regularization technique that constructs virtual training examples as:

$$\tilde{\mathbf{z}}_{\text{Mixup}} \equiv \lambda \mathbf{z}_i + (1 - \lambda)\mathbf{z}_j, \tag{6}$$

where $\mathbf{z}_i, y_i$ are the $i$-th input and label data, $\lambda \sim \text{Beta}(\alpha, \alpha)$ is a random scalar value controlling mixing ratio between the two samples, and $\alpha \in (0, \infty)$. MIXUP is typically applied across the elements of a given batch, and randomly produces new virtual samples by linearly mixing two classes as shown in Equation 6. While MIXUP was originally proposed to address problems such as reducing memorization of corrupted labels and increasing the robustness to adversarial examples, we observed limitations to its performance when we extended it to embedding methods (refer to Table 3). We hypothesize that the linear nature of MIXUP-generated example restricts the magnitude of gradients without changing their direction, which limits its effectiveness. On the other hand, EMU overcomes this limitation by producing updates that can take multiple directions and thus, enhances model training.

For the MIXUP experiments, we simply replaced the embedding mutation into Mixup in Equation 6. For simplicity, we set $\lambda \sim \text{Beta}(\alpha, \beta)|_{\alpha=2, \beta=1}$. Note that here we did not set as $\alpha = \beta$, as in the original implementation (Zhang et al., 2018), because we found that using different values of $\alpha$ and $\beta$ resulted in a significantly improved accuracy. We attribute this to the skewed probabilistic distribution that arises due to the different values of $\alpha$ and $\beta$, which allows for a higher ratio of negative samples than positive samples in the mixed-tail embedding vectors.

## H  T-SNE

In the main body of our study, we provide a scatter plot in Figure 5 to visualize the distribution of true and negative samples using the first and second components of PCA . Furthermore, to better capture the geometry in high-dimension, we also plot the distribution of h*r and negative-sample tails using tSNE in Figure 7. Our results show that EMU leads to a distribution that is more similar to the h*r than the uniform negative sampling case.

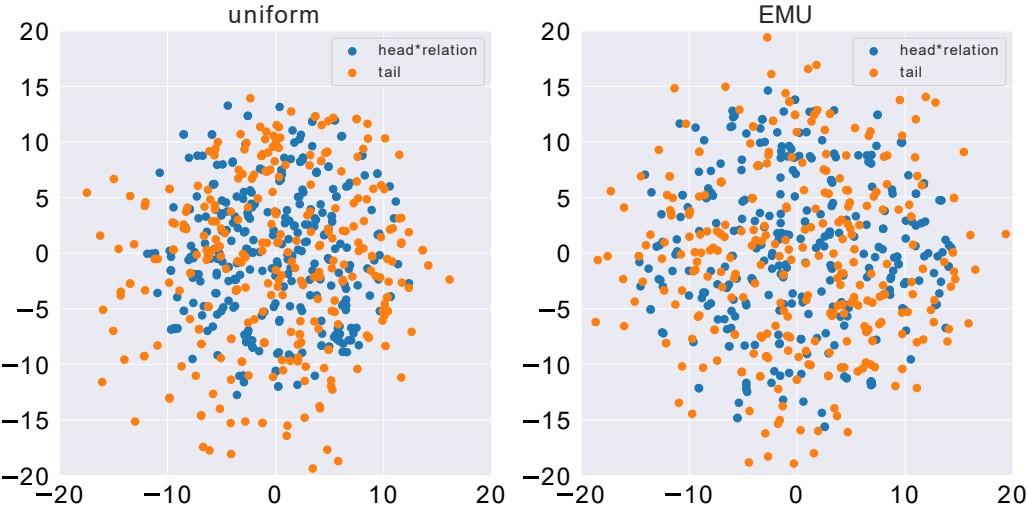

Figure 7: tSNE distribution of head * relation and tail for FB15k-237. The KG model is the Dist-Mult.

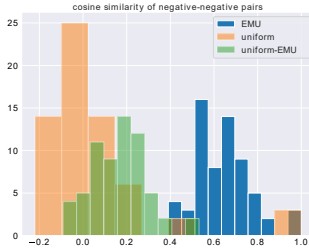

Figure 8: Cosine similarity of negative-negative tail pairs for DistMult with FB15k-237.

## I  EUCLIDEAN DISTANCE BETWEEN POSITIVE AND NEGATIVE SAMPLES GENERATED BY MIXUP AND EMU

The generation of new samples by EMU is expected to have a distinct impact on their position in the latent space. In particular, we can demonstrate that the distance between the newly generated example and a true sample is always equal to or lower than that between the original negative sample and a true sample because of the nature of EMU generating a harder negative sample. The vanilla Euclidean distance can be written as:

$$d_{\mathrm{PN}} = \sqrt{\frac{1}{d}\sum_{i=1}^{d}(\mathbf{z}_i^+ - \mathbf{z}_i^-)^2}. \tag{7}$$

While in the case of MIXUP, the distance is

$$d_{\mathrm{Mixup}} = \sqrt{\frac{1}{d}\sum_{i=1}^{d}(\mathbf{z}_i^+ - \mathbf{z}_i^{\mathrm{Mixup}})^2}$$
$$= (1-\lambda)d_{\mathrm{PN}} < d_{\mathrm{PN}}, \tag{8}$$

In the case of EMU, the distance is

$$d_{\mathrm{EMU}} = \sqrt{\frac{1}{d}\sum_{i=1}^{d}(\mathbf{z}_i^+ - \mathbf{z}_i^{\mathrm{EMU}})^2}$$
$$= \sqrt{\frac{1}{d}\sum_{i=n_{\mathrm{T}}+1}^{d}(\mathbf{z}_i^+ - \mathbf{z}_i^-)^2}$$
$$\sim \sqrt{1 - \frac{n_{\mathrm{P}}}{d}}\, d_{\mathrm{PN}}, \tag{9}$$

where in the last line, we approximated as $|\mathbf{z}_i^+ - \mathbf{z}_i^-| \sim d_{\mathrm{PN}}/d$ for the order estimation of the equation. In Equation 9 we assume the first $n_{\mathrm{P}}$ components in $\lambda_{\mathrm{EMU}}$ is unity and the others are zero: $\lambda_{\mathrm{EMU}} = \{1, 1, \cdots, 1, 0, \cdots, 0\}$, for simplicity.

The above equations show that EMU enables to generate hard negative samples similar to MIXUP but in a different manner.

## J  COSINE SIMILARITY BETWEEN NEGATIVE SAMPLES

Figure 8 plots the cosine similarity among negative samples for EMU-EMU, uniform-uniform, and uniform-EMU. The results indicate that the similarity between uniform negative samples are consistently lower than that of EMU negative samples, suggesting that EMU generates more hard negative samples.