# OpenReview forum: "EMU: EFFICIENT NEGATIVE SAMPLE GENERATION METHOD FOR KNOWLEDGE GRAPH LINK PREDICTION"
_ICLR.cc/2024/Conference — ICLR 2024 Conference Withdrawn Submission_

### Official Review · Reviewer_nmRt · 2023-10-26

**Soundness:** 2 fair
**Presentation:** 3 good
**Contribution:** 1 poor
**Rating:** 3
**Confidence:** 5

**Summary:**

This paper proposes a negative sampling method for knowledge graph link prediction. The key idea is to mix up the embeddings of the positive sample with several negative samples. To avoid the negative effect in penalizing the positive samples, an unbounded label smoothing method is proposed.

**Strengths:**

1. The idea is direct and easy to follow.
2. The presentation is smooth.

**Weaknesses:**

1. The scope of this paper is limited.
- Negative sampling is not essential in KG link prediction [1-2]. Cross entropy loss is also not essential [2]. But the proposed method is limited to the two constraints.
- Negative sampling itself is not limited to KG. It also plays important roles in general graph learning, like node2vec, or contrastive learning. Designing a method for a wider range can enhance the scope of this paper.
- The scoring functions studied in this paper are out of date.

2. There is no theoretical support on why this method can work. The design is mainly by empirical trials and intuitions. Many published negative sampling works were analyzed theoretically.

3. The results of embedding models are far from sufficiently trained.
- The performance of RotatE on FB15k-237 is 0.338 for MRR and 0.533 for Hit@10, and on WN18RR is 0.476 for MRR and 0.571 for Hit@10. However, the reported results in Figure 3 and Table 2 do not reach these values. It is hard to judge whether the performance gaining is from hyperparameter tuning or this method. I'm also not sure whether the performance still can be improved if the four models are fully tuned like by the training method in [2].
- The other datasets also face this problem, [2] as a paper in 2019 has better performance than the results reported in this paper.

4. Only two baselines are compared in this paper, namely SAN and NSCaching. What about the self-adversarial method in [3]? Based on my impression, there are several methods of studying negative sampling for KG.

5. No analysis or experimental results demonstrate the efficiency of this method.

6. WN18RR, WN18-RR, WN18rr.

[1] Li, Zelong, et al. "Efficient non-sampling knowledge graph embedding." Proceedings of the Web Conference 2021. 2021.
[2] Ruffinelli, Daniel, Samuel Broscheit, and Rainer Gemulla. "You can teach an old dog new tricks! on training knowledge graph embeddings." International Conference on Learning Representations. 2019.
[3] Sun, Zhiqing, et al. "RotatE: Knowledge Graph Embedding by Relational Rotation in Complex Space." International Conference on Learning Representations. 2018.

**Questions:**

1. Can you provide a theoretical analysis on why this method can work?
2. Can you explain why the results are far from SOTA? Or can you reproduce the results to reach the bar？
3. Can you compare with more baselines and more settings?

---

### Official Review · Reviewer_HCYP · 2023-10-31

**Soundness:** 2 fair
**Presentation:** 3 good
**Contribution:** 3 good
**Rating:** 5
**Confidence:** 4

**Summary:**

This paper proposes a newly-devised negative sampling technique that mixes the negative samples with the true samples to produce harder samples. The method is called EMU and it substitutes some elements of the negative samples with the ones of the true samples at some possibility. While the negative samples are crucial to learning the KG embedding space, the harder samples can effectively promote the experimental performance (MRR).

**Strengths:**

+ The technique seems to suit the intuition. The intuition is that harder negative samples are closer to true samples, so the model has to be trained to be more powerful to distinguish hard negative samples from true samples.
+ The approach is easy to accomplish and follow and I am sure that it can be incorporated into other KGE approaches easily.

**Weaknesses:**

- The authors claim that one characteristic of EMU is that it generates hard negative samples rather than identifying more difficult negative samples, but the necessity of generating new samples is unclear. Moreover, some GAN-based negative sampling techniques also have this characteristic that generating new samples. They are cited in the paper but not compared in the experiments, so it remains uncertain whether EMU can outperform other generative sampling approaches.
- As I understand, the authors want to let the negative samples closer to the true samples to make them `harder’, but it seems to just inject the noise into the true samples and regard the noisy true samples as newly-generated negative samples. I think this perspective is strongly related to noise-contrastive estimation (NCE), but the authors lack corresponding literature.
- The impact of the negative samples is not well introduced. The authors claim that the reason why negative samples are important is because KGs only contain positive links, which is insufficient and is not persuasive.
- The paper layout can be improved. For example, Figure 2 is on page 5, but the corresponding description is on page 3, which makes it hard to read them simultaneously.

**Questions:**

None

---

### Official Review · Reviewer_ZT2x · 2023-11-02

**Soundness:** 2 fair
**Presentation:** 2 fair
**Contribution:** 1 poor
**Rating:** 1
**Confidence:** 5

**Summary:**

Proposes a method to improve training of KGE models when negative sampling is used. The key idea is to augment the (embeddings of) negative samples using CutMix (termed EMU in paper) and using an "unbounded" form of label smoothing. An experimental study shows improvements over certain baselines.

**Strengths:**

S1. Simple, cheap, plug-in method. The method can be incorporated into any negative-sampling based KGE training pipeline at virtually no additional cost.

S2. Shows improvements over the baselines considered in the study.

**Weaknesses:**

W1. Low novelty. The particular augmentation method used in the paper is equivalent to CutMix (which the authors seem to be unaware of) applied to KGE, and "unbounded" label smoothing is a small trick/heuristic.

W2. Performance not convincing. First, the reported performance numbers produced in the study fall behind the known performance of the considered models on the datasets. For example, the paper reports ComplEx on FB15k-237 with an MRR 0.344 when trained with their method. This is lower than the study of Ruffinelli et al. cited in the submission (0.348) and lower than what can be achieved with ComplEx (0.37, [A]). It's also far behind the state-of-the-art (e.g., 0.415, [B]). One may argue that performance isn't everything, but a paper that aims to improve training methods should at least provide performance boosts when compared to known methods. This is emphasized by the fact that the experimental study apparently uses a VERY extensive hyperparameter search (10000 trials reported in appendix). Moreover, the study should use datasets that are so large that negative sampling is actually needed.

W3. No analysis/insight. The paper does not provide any insight or analysis into why improved performance is obtained. While there is prior evidence for CutMix, the proposed unbounded label smoothing needs more justification. At first glance, it appears unsound and it's not clear to me why it is a good idea, what the training process is optimizing with such an objective, and what it's properties are.

W4. Related work not adequately discussed. The paper does not give a good picture or related work on prior work on obtaining hard negatives. Many approaches are only mentioned in passing, MixUp is mentioned for the first time at the very end, and the last footnote remark is that MixUp has been applied to KGE training before.

[A] Lacroix et al., Canonical Tensor Decomposition for Knowledge Base Completion, ICML 2018
[B] Zu et al., Neural Bellman-Ford Networks: A General Graph Neural Network Framework for Link Prediction, NeuRIPS, 2021

**Questions:**

None

---

> ### Author Response · Authors · 2023-11-15
> **Comment on W1**
>
> As a message for future readers, we would like to differentiate between CutMix and EMU. CutMix works by blending two "input" images, creating new data by cropping a region from one image and replacing it with another. On the other hand, EMU merges positive and negative tails representations in the latent space.
> Addressing the second point, we fundamentally disagree with the reviewer's perspective. We believe it's a prime example of the "Egg of Columbus" and the comment lacks a coherent logical justification. It's also worth noting that Table 10 demonstrates that label-smoothing method hinders feature mutation from functioning correctly, resulting in drastically poor performance.

---

### Official Review · Reviewer_bmse · 2023-11-05

**Soundness:** 3 good
**Presentation:** 3 good
**Contribution:** 3 good
**Rating:** 6
**Confidence:** 4

**Summary:**

The paper describes a novel negative sampling approach (EMU) for the knowledge graph link prediction task. It is applicable to various KG link prediction algorithms (e.g DistMult, TransE), that rely on contrastive learning. The intuition behind the method is to produce negative examples using an analogy with gene mutations -- intuitivelly, the negative examples are created by 'mutating' the positive examples/vectors using the negative examples/vectors along some components. So, taking the Z_p and Z_n to be vectors corresponding to the positive and negative examples, the new 'mutated' negative sample is \lambda . Z_p  + (1 - \lambda) . Z_n, where \lambda is a binary valued vector denoting the 'mutated' vector components.
The authors also apply two versions of their sampling using the previously proposed label smoothing approach: EMU with and EMU without label smoothing, which allows for controlling the 'strength' of the penalty for the native class in the objective function (cross entropy loss, standardly used in the link prediction task). Finally, they compare their sampling technique using different link prediction algorithms and show that the performance of their sampling is on par or better than other sampling techniques.

**Strengths:**

Pros
+ Good paper articulating a somewhat simple but novel idea.
+ Very readable, provides good intuition and overview of the link prediction task, existing methods and the role that negative sampling plays in both.
+ The method is evaluated on the link prediction task using several datasets. Additionally, other experiments are provided to support the significance of the approach.

**Weaknesses:**

Cons

-  The experiments can be expanded by using more datasets, potentially including larger datasets (e.g. ConceptNet) or KGs from other domains (e.g. biological KG). This is not a strong con, as I understand the authors compare to other similar papers, which typically report performance on these NLP datasets.

- There are various abbreviations which are not defined. e.g.
    * MUTUP in the Limitations paragraph
    * SAN (Structure Aware Negative Sampling) is defined only in the Related work section but the abbreviation is used thoughout the paper.
    * similary, 'hard negative' is not defined (but used) until later pages in the paper.

- The Figures are bit hard to follow (see below)

**Questions:**

* While the paper is well written and provides intuitive explanations and a good literature review, the readability of the paper can be improved in terms of the abbreviations used (see above)

* Please move the Related work to earlier in the paper. As is, it is hard to understand what the baselines are until much later in the paper.   This would make the paper more easy to follow. I think this would also resolve all of the abbreviation issues listed above

* Please polish up the figures. Right now it is hard to understand if the colors in the positive/negative vectors matter in Fig 1(left)-- I don't think so, but if not, make that explicit in the caption.

* Fig 2. uses the notation \lambda_{Mutup}. What is that? The text uses \lambda_{EMU}. Please clarify or fix so the notation is consistent.

---

### Author Response · Authors · 2023-11-15
**General comment**

We thank the reviewers. Regrettably, based on the given scores, we've concluded that achieving acceptance for our paper in its present state is virtually impossible. Therefore, we've decided to withdraw it. However, we still firmly believe in the value of our work.
For future readers, we've addressed a few common criticisms received from the reviewers:

Baseline Score: The baseline score is partly influenced by the standardization of our training setup, which involves the use of common negative sample numbers, among other things. We also discovered that model performance can vary even when the same hyper-parameter values are used. While the latter issue is challenging to rectify, we plan to enhance the baseline model performance through a more extensive hyperparameter tuning.

Analysis: We've included an analysis of the impact of feature Mutation and unbounded label smoothing in Sections 4.4, 4.5, and in the supplementary material. We aim to address theoretical analysis in our next submission.